# Ultracold atom interferometry in space

Maike D. Lachmann [1,16], Holger Ahlers[1,16], Dennis Becker[1], Aline N. Dinkelaker [2,12], Jens Grosse[3,4], Ortwin Hellmig[5], Hauke Müntinga [3,13], Vladimir Schkolnik [2], Stephan T. Seidel[1,14], Thijs Wendrich[1], André Wenzlawski[6], Benjamin Carrick[7,15], Naceur Gaaloul[1], Daniel Lüdtke [7], Claus Braxmaier[3,4], Wolfgang Ertmer[1], Markus Krutzik[2], Claus Lämmerzahl[3], Achim Peters[2], Wolfgang P. Schleich [8,9,10], Klaus Sengstock[5], Andreas Wicht[11], Patrick Windpassinger[6] & Ernst M. Rasel[1✉]

Bose-Einstein condensates (BECs) in free fall constitute a promising source for space-borne interferometry. Indeed, BECs enjoy a slowly expanding wave function, display a large spatial coherence and can be engineered and probed by optical techniques. Here we explore matter-wave fringes of multiple spinor components of a BEC released in free fall employing light-pulses to drive Bragg processes and induce phase imprinting on a sounding rocket. The prevailing microgravity played a crucial role in the observation of these interferences which not only reveal the spatial coherence of the condensates but also allow us to measure differential forces. Our work marks the beginning of matter-wave interferometry in space with future applications in fundamental physics, navigation and earth observation.

[1] Institute of Quantum Optics and QUEST-Leibniz Research School, Leibniz University Hannover, Hannover, Germany. [2] Department of Physics, Humboldt-Universität zu Berlin, Berlin, Germany. [3] Center of Applied Space Technology and Microgravity (ZARM), University of Bremen, Bremen, Germany. [4] Department of Enabling Technologies, German Aerospace Center (DLR), Bremen, Germany. [5] Institute of Laser-Physics, University Hamburg, Hamburg, Germany. [6] Institute of Physics, Johannes Gutenberg University Mainz (JGU), Mainz, Germany. [7] Institute for Software Technology, German Aerospace Center (DLR), Brunswick, Germany. [8] Institut für Quantenphysik and Center for Integrated Quantum Science and Technology (IQST), Ulm, Germany. [9] Institute of Quantum Technologies, German Aerospace Center (DLR), Ulm, Germany. [10] Hagler Institute for Advanced Study at Texas A&M University; Texas A&M AgriLife Research; Institute for Quantum Science and Engineering (IQSE) and Department of Physics and Astronomy, Texas A&M University, College Station, TX, USA. [11] Ferdinand-Braun-Institut, Leibniz-Institut für Höchstfrequenztechnik, Berlin, Germany. [12] Present address: Leibniz-Institut für Astrophysik Potsdam, Potsdam, Germany. [13] Present address: Institute for Satellite Geodesy and Inertial Sensing, German Aerospace Center (DLR), Bremen, Germany. [14] Present address: Airbus Defense and Space GmbH, Taufkirchen, Germany. [15] Present address: MORABA, German Aerospace Center (DLR), Weßling, Germany. [16] These authors contributed equally: Maike D. Lachmann, Holger Ahlers. ✉email: rasel@iqo.uni-hannover.de

nterference of two BECs constitutes the hallmark of macroscopic coherence. The first observation[1] of the corresponding fringes has ushered in the new era of coherent matter-wave optics.

The condensates can be engineered and probed by optical techniques[2–5] making them a promising source for precision measurements. In addition, their large spatial coherence and their slow expanding wave function[6] allow for experiments on macroscopic time scales.

Indeed, space displays an enormous potential for advancing high-precision matter-wave interferometry because the size of the device is no longer determined by the dropping height required on ground[7]. Additionally, low external influences and the reduced kinematics of the source allow us to control systematic effects. For these reasons quantum tests of general relativity[8–10], the search for the nature of dark matter and energy[9,11–14], the detection of gravitational waves[9,15,16] and satellite gravimetry[17–21] represent only a few of the many promising applications of atom interferometry in space.

Being at the very heart of the aforementioned proposals, our experiments set the beginning of space-borne coherent atom optics. They have benefited from our earlier studies on BEC interferometry at the drop tower in Bremen[22,23] exploring methods for high-precision inertial measurements. Moreover, in other groups interferometry with laser-cooled atoms was performed on parabolic flights[24] and a cold-atom clock was studied on a satellite[25]. Exploration of degenerate quantum gases is currently continued with the Cold Atom Laboratory (CAL) in orbit[26].

In this article we report on the first interference experiments performed during the recent space flight of the MAIUS-1 rocket[27] demonstrating the macroscopic coherence of the BECs engineered in this microgravity environment.

## Results

In contrast to[1] which used an optical double-well potential to interfere two BECs, we employ Bragg processes as well as phase imprinting simultaneously in different magnetic spinor components. The resulting rich interference pattern stretches across the complete spatial distribution of the wave packets. We analyse its contrast as its temporal evolution can be exploited to detect forces acting differently on the spinor components. Applied to three dimensions, this arrangement could serve as a vector magnetometer. Moreover, due to the higher achievable spatial resolution, our method compares favourably to the determination of the BEC position based on a fit of the envelope of its spatial density distribution. The latter was e.g. proposed for the measurement of gravity gradients in the STE-QUEST[8] mission designed as a space-borne quantum test of the equivalence principle.

**Setup**. Our interferometer is based on an atom-chip apparatus for trapping and cooling of the atomic ensembles delivering BECs of about $10^5$ rubidium-87 atoms within 1.6 seconds[28,29]. The atom chip enables a very compact and robust design required by the demanding constraints of the rocket in terms of mass, volume, power consumption as well as vibrations and accelerations during launch. The necessary autonomous experimental control is realised by a customised onboard software allowing for image analysis, self-optimisation of parameters, operation of a decision tree and real time interaction with the ground control.

The BECs are created in the magnetic hyperfine ground state F = 2, $m_F$ = +2 and then released from the trap. We transfer the freely falling matter-wave packet into a superposition of several spinor components by employing radio frequencies[30], or by changing the magnetic field orientation. Optionally, we spatially separate them after the interferometry sequence by a Stern–Gerlach arrangement.

The experiments reported here were performed with a superposition of atoms in the three spinor components $m_F$ = ±1 and 0, which are synchronously and identically irradiated by a single or sequential light pulses creating spatial matter-wave interferences. Figure 1a depicts the arrangement of the light fields, which are detuned by 845 MHz with respect to the D2 transition in rubidium, and drive several processes detailed in Fig. 1b–d.

Perpendicular to the direction of absorption imaging (green circle) along z, two counterpropagating light beams A and B run parallel to the atom chip, and induce Bragg diffraction in the x-direction such that the diffracted wave packets gain two photon recoils in momentum. The direction of momentum transfer is tilted with respect to the longitudinal axis of the sounding rocket, which is oriented along the diagonal of the x–z plane. This configuration is chosen such that in space, the separation points along Earth's gravitational pull g, and on ground, at an angle of 45° to the latter.

Bragg diffraction (Fig. 1b) is resonantly driven by tuning the frequency difference, $\nu_A - \nu_B$, between the two light beams to create an optical lattice travelling with half of the speed of the diffracted wave packets. Timing and duration as well as frequency difference and power adjustment are performed autonomously via acousto-optical modulators during space flight.

Reflections of the Bragg beams on the optical viewports of the vacuum chamber create additional low-intensity light beams with a tilt of about two degrees determined by comparing experiments and simulations. They interfere with the original beams A and B giving rise to an additional spatial intensity modulation moving approximately along the y-direction. This effect leads to an averaged phase imprinting on the wave function[2] (Fig. 1c) as well as to Bragg double-diffraction of the wave packets, which is comparably weak due to the low intensities of the reflections. In addition, the light beams A and B are diffracted at the edges of the atom chip, and hence, feature a spatial intensity modulation, which also causes phase imprinting roughly oriented along the y-direction (Fig. 1d).

For a systematic analysis we implemented numerical simulations of our experiments. The theoretical images of the spatial density distributions depict simulations modelling the evolution of the different spinor components of the wave packets including the atom-light interaction in position space[31,32]. Parameters, such as relative intensities, frequencies and the geometry, are independently determined by the experimental setup. The 2° beam angle and an overall intensity adjustment were adapted.

**Coherent manipulation by light pulses**. The key processes of our interference experiments can be understood by analysing the effect of a single light-pulse on the multi-spinor BECs. Figure 2 compares the experiments in space (left column) and on ground (right column), in which the light fields interact for 60 µs with a multi-component wave packet 15 ms after its release. Moreover, the experimental results obtained with multiple spinor components are contrasted with the corresponding simulations, which serve as a reference and consider a single wave packet in the state F = 2, $m_F$ = 0.

The figure shows the spatial density distribution of the wave packets and its Bragg diffracted parts. The experimental images were obtained by absorption imaging 31 ms and 86 ms after release on ground and during the rocket flight, respectively.

Most strikingly, the results obtained in space feature a pronounced horizontal stripe pattern, which is oriented almost along the y-direction with a period of roughly 60 µm. In contrast, the pictures on ground do not display such a pattern.

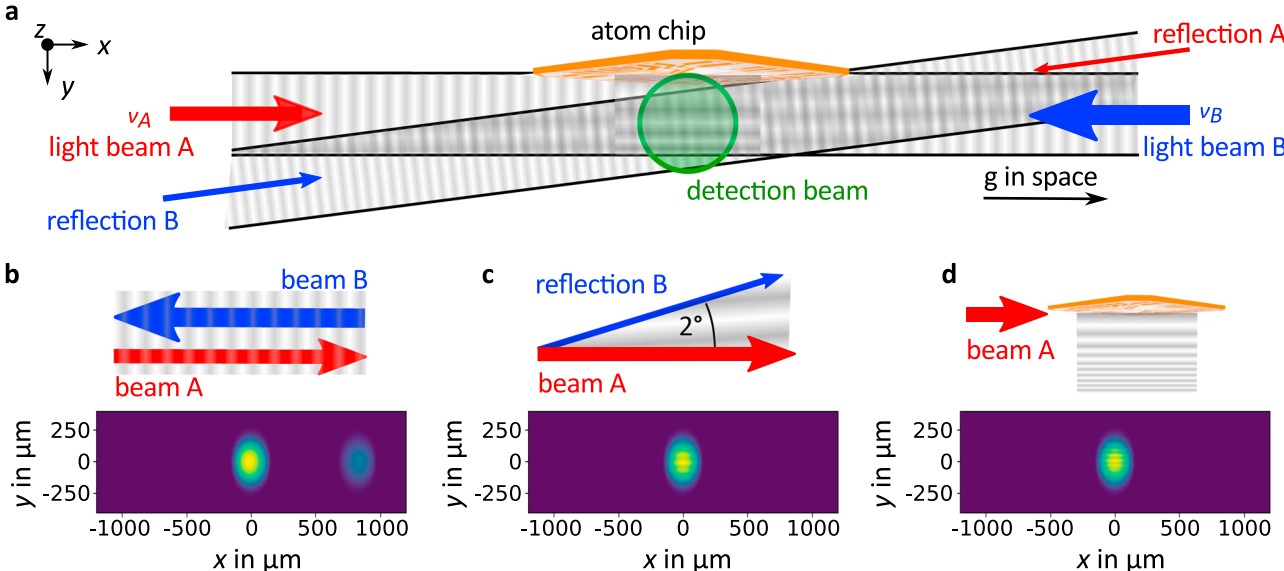

**Fig. 1 Optical setup.** Optical arrangement for space-borne light-pulse interferometry employing a BEC and associated diffraction processes. **a, b** After release of the multi-component rubidium BEC two light beams, A and B, with different frequencies $\nu_A$ and $\nu_B$, and intensities travel in opposite directions parallel to the atom chip and generate a moving optical lattice driving Bragg processes, which coherently transfer momentum to the atomic wave packet along the x-direction (**b**). Two additional light beams tilted by two degrees emerge due to reflections of the beams A and B on the optical viewports. **c** Their interference with the lattice beams gives rise to a traveling spatial intensity modulation in the y-direction modifying the BEC wave function as well as inducing weak double-Bragg processes in x-direction (not shown). **d** In addition, the light beams are diffracted at the atom chip and the arising interference modulates their intensity in y-direction. The various effects of the light pulses on the multi-component wave packet are detected by a CCD-camera recording the shadow of the BEC irradiated by light (green circle) from the z-direction. Earth gravity pulls along the x-direction during space flight, and along the x–z diagonal on ground.

We identify four reasons for this clear distinction between ground and space experiments: (i) In order to cope with the gravitational pull on ground, a time of flight shorter than in space had to be chosen as to ensure that the wave packet is detected close to the focal plane of the imaging. (ii) The corresponding shorter expansion time leads to a smaller size of the wave packet, and the fringe spacing imprinted onto the wave packet by the diffraction of the Bragg beams at the edges of the atom chip is below the image resolution. (iii) On ground, the frequencies of the Bragg beams were adjusted to compensate the projection of the gravitational pull. Therefore, the interference pattern of the reflected and incoming light fields moves with a larger speed than in space and wash out the related phase imprint. (iv) Double-Bragg process are suppressed as they are non-resonant.

However, in the microgravity environment of the sounding rocket, the frequency difference of the light beams tuned to the Bragg resonance results in an optical interference pattern travelling with a lower speed and a temporal periodicity close to the recoil frequency. This motion is slow enough to leave a phase imprint albeit with lower amplitude due to temporal averaging over the 60 μs of interaction. Our simulations of the experiments with and without gravity confirm this difference and allow us to deduce the reflection angle from the fringe spacing.

In our space experiments the observed fringe contrast is still much lower than predicted by our simulations based on a single spinor component. This deviation becomes even more prominent when we consider a slice through the intensity modulation along the y-direction indicated by the orange line in Fig. 2. The existence of several spinor components in presence of a residual magnetic field gradient explains this reduction. Indeed, the latter suffices to accelerate the individual components relative to each other according to their different magnetic susceptibilities. Since our imaging does not distinguish between the different components we arrive at a lower contrast.

**Interference experiments**. This assumption is confirmed by the experiments summarised in Fig. 3. Here we study interferences generated by three sequential light pulses acting synchronously and identically on all spinor components. Moreover, we perform a Stern–Gerlach analysis of the interferometer output ports, which can be clearly distinguished by their kinetic momenta due to the use of BECs as a source.

Figure 3a depicts the interferometric arrangement to coherently split, deflect and recombine the different spinor components leading to a grid-like stripe pattern detailed in Fig. 1b–d. The sequence of pulses is reminiscent of a Mach–Zehnder-type interferometer but our Bragg processes were weak and a momentum transfer occurred only to a small fraction of a BEC.

The pictures were taken 50 ms after exposure of the released BECs to the magnetic field gradient for state separation, and 67 ms after the third light-pulse. This choice of parameters guaranteed that the exit ports were spatially separated on the absorption images according to their momenta and spinor components at as indicated by the red lines in Fig. 3a.

Moreover, the time between the light pulses was 1 or 2 ms and, hence short enough, that the wave packets largely overlap at the exit ports giving rise to interference fringes modulated approximately along the x-direction. Therefore, the experimental arrangement resembles a shearing interferometer probing the spatial coherence of the different spinor components.

Figure 3b, c depicts the enlarged view of the output port corresponding to the $m_F = -1$ state together with the line integral along a fringe, and our theoretical simulations of both, respectively. Indeed, the pattern analysis shares similarities with point-source interferometry[23,33]. While theory and experiment feature the same spatial periodicity, we had to add an inhomogeneous field in our simulation to obtain agreement of the tilts of the fringe pattern. Such a residual field is also required to explain the orientation of the phase imprint discussed below.

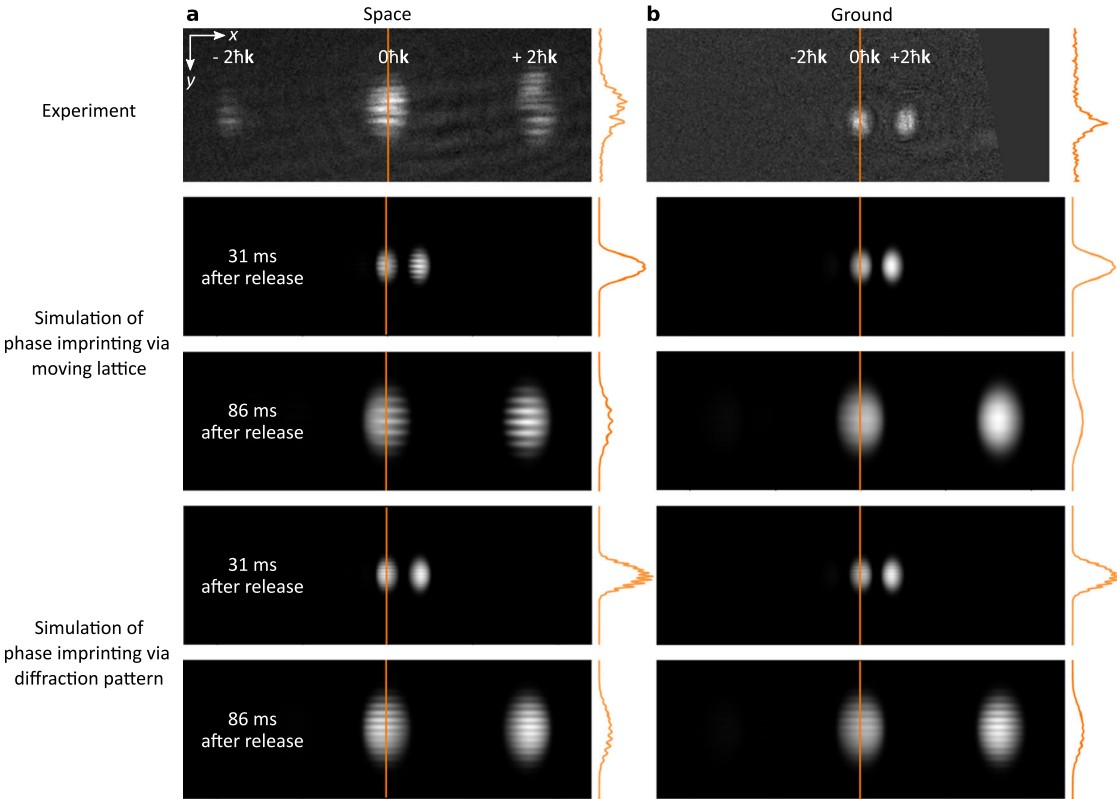

**Fig. 2 Effects of a single light-pulse.** Depicted are the experimental observations (two upper panels) and theoretical simulations (lower panels) of the impact of a single light-pulse simultaneously inducing Bragg processes and phase imprinting on a matter-wave packet released from the trap in the low gravity environment of space (**a**) (left column), and on ground (**b**) (right column). Each picture shows the undiffracted and the diffracted parts due to Bragg and weak double-Bragg processes. In space, the striking feature is the amplitude modulation of all Bose-Einstein condensates along the y-direction which in our ground experiments was not visible, in accordance with our theoretical simulations. On ground, the free expansion time of the BECs was short (31 ms) to keep them in the Bragg beams and the focal region of the detection. In space, the longer expansion times (86 ms) lead to a larger fringe spacing which allows us to resolve the imprint. In addition, the phase imprinting due to the moving amplitude modulation vanishes on ground due to the Doppler shift caused by the larger detuning between the light beams A and B, resulting in a much larger velocity of the light pattern. In contrast to our model assuming a single BEC component, the experimental fringe patterns feature spatial distortions as well as a much lower contrast. The latter holds even in the case when only a segment of the picture along the y-direction (orange line) is analysed.

The low contrast observed in the experiment, of about 20% in the line integral, can be explained by inhomogeneities of the light beams used for Bragg diffraction leading to a spread of Rabi frequencies. In these experiments we have benefited from the point-source character of the BEC as our theoretical simulation reveals that the spatial interference pattern would vanish already for a thermal cloud of atoms with temperatures of a few hundred nK.

The interaction of the BEC with the three light fields also leads to phase imprinting, and hence, to a notable stripe pattern along the y-direction as confirmed by our simulations.

According to our theory such a pattern can originate from a repetitive imprint by light diffracted at the chip edge as discussed in Fig. 1d. Even more remarkably and despite the averaging, our theory also reveals that, for our optical arrangement depicted in Fig. 1a, the moving amplitude modulation detailed in Fig. 1c, leads to a phase imprint featuring the observed fringe spacing.

Without the Stern–Gerlach analysis and therefore spatially overlapping spinor components, the absorption images and simulations of the patterns exemplified by Fig. 3d, e, show a low contrast. In comparison, the Stern–Gerlach separation leads to a higher fringe contrast, and allows us to selectively visualise the fringe patterns for the different spinor components as illustrated in Fig. 3f.

In order to separate the effect of the fringe tilts and inhomogeneities we restrict our analysis to vertical segments along the orange line. Indeed, the patterns corresponding to $m_F = \pm 1$ are rotated in opposite direction which can be explained by a magnetic field curvature. While a homogeneous magnetic field gradient would just lead to a translation of the $m_F = \pm 1$ components with respect to the $m_F = 0$ component, a curvature induces tilts. Our simulations shown in Fig. 3g–i confirm this effect and feature corresponding patterns for a value of 3.5 μT/mm² for the curvature of the magnetic field.

## Discussion

Hence, simultaneous imprinting of a stripe pattern onto a multi-component or multi-species wave packet formed from a BEC using a spatially modulated far-detuned light beam allows us to analyse differential forces due to external electric or magnetic field gradients and curvatures, or to detect a differential velocity of the components. While a pure translation can be observed by the resulting loss of contrast, the detection of tilts would preferably be combined with a Stern–Gerlach separation.

To detect forces by tracking the motion of the center-of-mass of the wave packets is a frequently used technique. For example, it was exploited in drop tower experiments[22], is studied with CAL[26], and proposed for STE-QUEST[8] to characterise the environment and movement of the atomic ensemble. Our method improves the sensitivity for wave packet displacements by fitting the smaller spatial fringe period instead of the envelopes. The

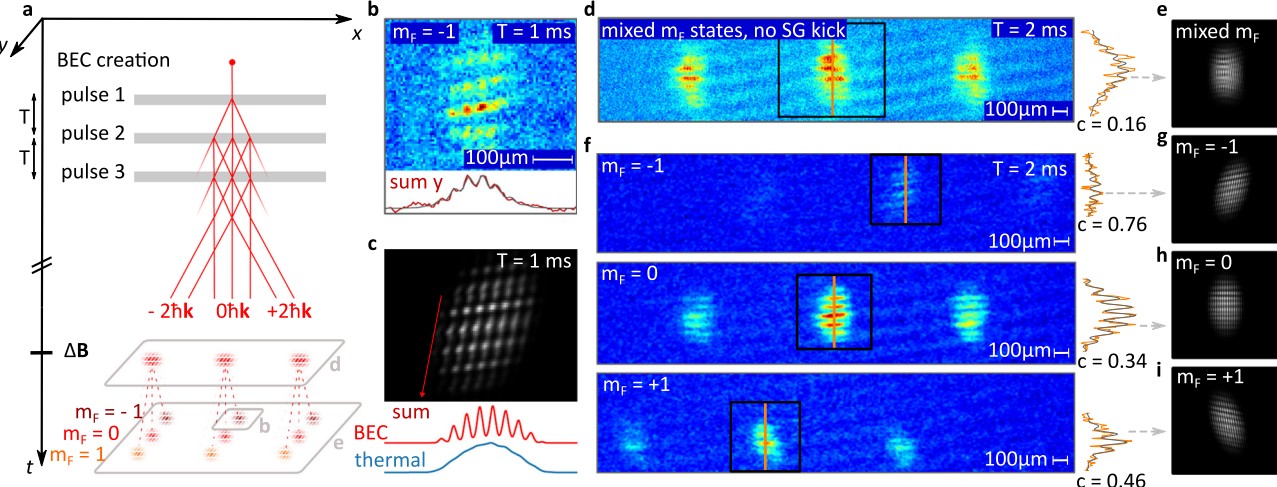

**Fig. 3 Experimental and simulated spatial matter-wave fringes.** The interference is created in a multi-component BEC by a sequence of three light pulses applied after release. **a** The associated Bragg processes create several spatially displaced, but still largely overlapping wave packets, resulting in an interference pattern in the three output ports of the interferometer corresponding to a transfer of either +1, −1 or 0 effective photon recoils. The fringes are recorded with and without a prior Stern–Gerlach-type spatial separation of the different spinor components. We model the experiment by solving the 2D-Gross-Pitaevskii equation of a BEC interacting with the light fields discussed in Fig. 1. **b**, **c** A close-up of one output port is shown with the corresponding line integrals along the red line (bottom) as well as their theoretical counterparts. The experiment displays a lower contrast due to spatially varying Rabi frequencies. The temporal sequence of the three light pulses also leads to an effective phase imprinting. **d–i** The stripe pattern (left) and contrast for a data slice (right) observed with and without the Stern–Gerlach separator are depicted. Without Stern–Gerlach separation the stripe pattern obtained for the slice along the orange line (**d**) features a lower contrast than our model (**e**) which might result from the relative motion of the spinor components. We observe a higher contrast for different magnetic states (**f**). Indeed, the components $m_F = \pm1$ feature a tilt of opposite sign with respect to the component $m_F = 0$ which points to a residual magnetic field with a curvature in agreement with our numerical simulations (**g–i**).

principle of using an interference modulation in the signal is reminiscent of the increased resolution in the Michelson stellar interferometer[34].

We foresee several extensions of the imprint method: (i) adaption to gravity in order to avoid loss of modulation depth due to the moving grating, (ii) application to three dimensions by spectral spatial light modulators for a 3D imprint and (iii) measurements of inertial forces acting on a single species or multiple ones in cases where Mach–Zehnder interferometers are not available, or their dynamic measurement range is surpassed.

In conclusion, we have employed light-pulse interferometry induced by Bragg processes as well as phase imprinting to investigate and exploit the spatial coherence of multi-component BECs on a sounding rocket. Our experiments mark the beginning of matter-wave interferometry in space and lay the groundwork for future in-orbit interferometry performed with CAL and its future successor BECCAL[35], for the next MAIUS missions, and generally, for high-precision interferometry in space.

## Methods

The optical setup for interferometry consists of two collimated and counter-propagating light beams A and B as well as their reflections with an angle of two degrees with respect to A and B as shown in Fig. 1a. Their frequencies are detuned by 845 MHz from the Rubidium-87 D2 line. To fulfil the Bragg condition, both beams need a relative detuning which at the beginning of the flight was set to $\nu_A - \nu_B = 15.1$ kHz. Unfortunately, a residual movement of the atoms after release from the magnetic trap reduced the diffraction efficiency. Therefore, it was adjusted during flight to a value of 18.8 kHz for later measurements.

The light pulse has a length of 60 μs and beam A an intensity of 4.1 mW/cm² while the beam B has an intensity of 8.0 mW/cm² for the analysis of single interactions. In the three-pulse sequences the intensity of beam A remains the same, whereas the intensity for beam B is doubled for the second pulse leading to an increased diffraction ratio. Approximately 5% of both light beams are reflected on the optical viewports of the vacuum chamber.

On ground, the interferometry axis is tilted by 45° with respect to gravity. In free fall the Doppler shift leads to a detuning of $\nu_A - \nu_B = 259.4$ kHz. For this reason, double diffraction is suppressed.

## Data availability

The image data for Figs. 2 and 3 are provided under https://doi.org/10.25835/0062691. The analysed data are available from the corresponding authors on reasonable request.

## Code availability

The simulation code is available from the corresponding authors on reasonable request.

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

## Acknowledgements
We thank all members of the QUANTUS-collaboration for their support and acknowledge fruitful discussions with M. Cornelius, P. Stromberger and W. Herr. This work is supported by the DLR Space Administration with funds provided by the Federal Ministry for Economic Affairs and Energy (BMWi) under grant numbers DLR 50WM1131-1137, 50WM0940, 50WM1240, 50WM1556, 50WM1641, 50WM1861, 50WM1956, 50WP1431-1435 and 50WM2060, and is funded by the Deutsche Forschungsgemeinschaft (DFG, German Research Foundation) under Germany's Excellence Strategy—EXC-2123 QuantumFrontiers—390837967. W.P.S. thanks Texas A&M University for a Faculty Fellowship at the Hagler Institute for Advanced Study at Texas A&M University and Texas A&M AgriLife for support of this work. The research of the IQ<sup>ST</sup> is financed partially by the Ministry of Science, Research and Arts Baden-Württemberg. H.A. acknowledges financial support from "Niedersächsisches Vorab" through "Förderung von Wissenschaft und Technik in Forschung und Lehre" for the initial funding of research in the new DLR-SI Institute. N.G. acknowledges funding from "Niedersächsisches Vorab" through the Quantum- and Nano-Metrology (QUANOMET) initiative within the project QT3. We thank ESRANGE Kiruna and DLR MORABA Oberpfaffenhofen for assistance during the test and launch campaign.

## Author contributions
M.D.L., H.A., D.B., A.N.D., J.G., O.H., H.M., V.S., T.W., A.We. and B.W., with S.T.S. as scientific lead, planned and executed the campaign. M.D.L. and H.A. evaluated the data. H.A. and N.G. carried out the simulations. E.M.R., W.P.S., M.D.L. and H.A. wrote the manuscript, with contributions from all authors. C.B., W.E., M.K., C.L., D.L., A.P., W.P.S., K.S., A.Wi. and P.W. are the co-principal investigators of the project, and E.M.R. its principal investigator.

## Funding

## Competing interests
The authors declare no competing interests.
