## [Peer Review File · Nature Communications]

Reviewers' Comments:

Reviewer #2:

Remarks to the Author:

This article reports the observations of interference due to coherence of atom waves, Bose Einstein Condensate, in the experiments that were carried out in 2017 MAIUS-1 sound rocket launch [Ref.1]. I found the manuscript clearly presents the main results and points. The major claims are also clearly substantiated in the manuscript.

In essence, the authors have done an BEC coherence experiments in a sounding socket and were able to simulate and explain the observed fringes and patterns resulted from the applied Bragg beams as well as unintended environment effects such as diffractions of the laser beams from the atom chip and residual magnetic field gradients. In addition to the fact that the experiments were carried out in space on a sounding rocket, they also observed higher fringe contrasts in the sounding rocket than those on the ground experiments, owing to the unique micro gravity environment in the free fall sounding socket. While I do not think the advantages of "space" (i.e. micro gravity) shown in the results are scientifically significant, it is precisely because of the need to compensate the gravity pull in the Bragg beams that makes the fringes in the ground experiments less visible, among several other minor reasons, and therefore, justifying the manuscript's main claim. The values of the results reported here have more to do with the accomplishments of carrying out the experiments in a sounding socket, which are indeed a pioneering and challenging work towards future atom wave interferometers in space, and the observed differences of carrying out such experiments on ground and in space. Therefore, I'd recommend the manuscript be published in Nature Communications, provided the following comments are taken into consideration before publishing:

- 1) In the abstract, "Our work 'establishes' matter-wave interferometry in space with future applications in fundamental physics, navigation and earth observation". I find "establish" way a strong word, and not justified here. I'd suggest it change to something like "mark the beginning" as in the main texts of the manuscript.
- 2) There is fair amount of discussions about using fringe motions as a higher resolution way to detect forces (Lines 52-55, 192-195). It is not clear what it compared to. It cited "For example, it was exploited in drop tower experiments²⁴, is studied with CAL²⁷, and proposed for STE-QUEST⁹.", so it leaves general readers to figure out. I'd suggest authors spell it out specifically. My guess is that the cited way is simply watching the direct BEC packet itself (authors mentioned the envelope of BEC). If so, I'd not believe is a truly practical measurement scheme. The comparison is not clear, and perhaps not worthwhile.
- 3) Line 100 "double Bragg diffraction", I am not sure authors meant simultaneous two diffraction processes, rather than double-diffraction which would mean the diffracted wave packets get diffracted one more time. Needs clarified.
- 4) Line 94, "is performed" should be "are performed", and perhaps a few typos in other places.

Reviewer #3:

Remarks to the Author:

I would like to thank the authors for their thoughtful answers and their work to improve the quality of

the manuscript.

MAIUS-1 is undoubtedly a project of main importance in the field of atomic physics for space applications and the outputs of such experiment should be of great interest for the community. Therefore, I believe that the presented results should be published in a high impact journal such as Nature Communications. Congratulations to the MAIUS team for all this ambitious work.

Nevertheless, a small remark remains for a better understanding of all the conducted work. I strongly suggest to add in the body of the manuscript (not only in the "Methods" part), line 96, that the reflection of the Bragg beams comes from the optical viewport. Otherwise, it raises too many questions during the reading. "Reflections of the Bragg beams on the optical viewports of the vacuum chamber create additional low-intensity" or something similar would be fine for me.

Referee 2:

We thank the referee for her/his recommendation to publish our manuscript and the comments to further improve it. The suggestions of the referee are individually addressed in the subsequent list. Here we first quote report in italics the relevant excerpt from the report followed by our response.

1) In the abstract, "Our work 'establishes' matter-wave interferometry in space with future applications in fundamental physics, navigation and earth observation". I find "establish" way a strong word, and not justified here. I'd suggest it change to something like "mark the beginning" as in the main texts of the manuscript.

We agree and changed the phrasing to:

“Our work marks the beginning of matter-wave interferometry in space with future applications in fundamental physics, navigation and earth observation.”

2) There is fair amount of discussions about using fringe motions as a higher resolution way to detect forces (Lines 52-55, 192-195). It is not clear what it compared to. It cited "For example, it was exploited in drop tower experiments²⁴, is studied with CAL²⁷, and proposed for STE-QUEST⁹.", so it leaves general readers to figure out. I'd suggest authors spell it out specifically. My guess is that the cited way is simply watching the direct BEC packet itself (authors mentioned the envelope of BEC). If so, I'd not believe is a truly practical measurement scheme. The comparison is not clear, and perhaps not worthwhile.

Indeed, we discuss about the method to calculate external forces by tracking the centre-of-mass motion of the wave package. For example, this method is on reasonable timescales of milliseconds to seconds used to characterise magnetic background fields by analysing the force and its effect on the motion of the matter wave. This has been done by the QUANTUS-I apparatus in the drop tower [1] and in CAL onboard the ISS [2]. In addition, tracking is used as a characterisation method for BEC release in the STE-QUEST proposal [3].

The determination of the centre-of-mass position is limited by shot-noise and uncertainties of the image analysis and fitting routine. For a smaller object, like the single stripe structures, the determination becomes more precise.

In order to clarify the statement, we changed the text to

“To detect forces by tracking the motion of the centre-of-mass of the wave packets is a frequently used technique. For example, it was exploited in drop tower experiments²⁴, is studied with CAL²⁷, and proposed for STE-QUEST⁹ to characterise the environment and movement of the atomic ensemble. “

[1] van Zoest, T., et al. Bose-Einstein Condensation in Microgravity. *Science*. **328**, 1540-1543 (2010).

[2] Aveline, D. C., et al. Observation of Bose–Einstein condensates in an Earth-orbiting research lab. *Nature*. **582**, 193-197 (2020).

[3] Aguilera, D. N., et al. STE-QUEST test of the universality of free fall using cold atom interferometry. *Classical and Quantum Gravity*. **31**, 115010 (2014).

3) Line 100 "double Bragg diffraction", I am not sure authors meant simultaneous two diffraction processes, rather than double-diffraction which would mean the diffracted wave packets get diffracted one more time. Needs clarified.

In our setup the two counterpropagating light beams with detuned frequencies for Bragg diffraction are reflected. As a result, we have two moving optical lattices in both directions, whereas the one of the reflected beams has a lower intensity. Due to the relatively small movement of the atoms in this axis in microgravity the Bragg condition is fulfilled in both directions. And therefore, simultaneous two-photon transitions in both directions appear. This process is called double-diffraction in the literature [3,4]. The difference in intensity leads to unequal diffraction efficiencies.

We change the formulation to: "Bragg double-diffraction".

[3] Ahlers, H. et al. Double Bragg Interferometry. *Phys. Rev. Lett.* 116, 173601 (2016)

[4] Hartmann, S. et al. Regimes of atomic diffraction: Raman versus Bragg diffraction in retroreflective geometries. *Phys. Rev. A* 101, 053610 (2020)

4) Line 94, "is performed" should be "are performed", and perhaps a few typos in other places.

We thank the referee for this helpful comment. We corrected the formulation and checked again for typos.

Referee 3:

We thank the referee for her/his recommendation to publish our manuscript in *Nature communications*. The last suggestion of the referee is addressed below. Here we first quote the report followed by our response.

Nevertheless, a small remark remains for a better understanding of all the conducted work. I strongly suggest to add in the body of the manuscript (not only in the "Methods" part), line 96, that the reflection of the Bragg beams comes from the optical viewport. Otherwise, it raises too many questions during the reading. "Reflections of the Bragg beams on the optical viewports of the vacuum chamber create additional low-intensity" or something similar would be fine for me.

Thanks for this suggestion. To avoid confusion, we added the information about the reflections now directly in the text:

"Reflections of the Bragg beams on the optical viewports of the vacuum chamber create additional low-intensity light beams with a tilt of about two degrees determined by comparing experiments and simulations."